# Advances and Challenges in Modeling Autosomal Dominant Polycystic Kidney Disease: A Focus on Kidney Organoids

**DOI:** 10.3390/biomedicines13020523

**Published:** 2025-02-19

**Authors:** Jinglan Gu, Fei Liu, Lu Li, Jianhua Mao

**Affiliations:** Department of Nephrology, Children’s Hospital, Zhejiang University School of Medicine, National Clinical Research Center for Child Health, Hangzhou 310058, China; gujinglan@zju.edu.cn (J.G.); feiliu81@zju.edu.cn (F.L.); luli1988@zju.edu.cn (L.L.)

**Keywords:** polycystic kidney disease, organoid, induced pluripotent stem cells

## Abstract

Autosomal dominant polycystic kidney disease (ADPKD) is a prevalent hereditary disorder characterized by distinct phenotypic variability that has posed challenges for advancing in-depth research. Recent advancements in kidney organoid construction technologies have enabled researchers to simulate kidney development and create simplified in vitro experimental environments, allowing for more direct observation of how genetic mutations drive pathological phenotypes and disrupt physiological functions. Emerging technologies, such as microfluidic bioreactor culture systems and single-cell transcriptomics, have further supported the development of complex ADPKD organoids, offering robust models for exploring disease mechanisms and facilitating drug discovery. Nevertheless, significant challenges remain in constructing more accurate ADPKD disease models. This review will summarize recent advances in ADPKD organoid construction, focusing on the limitations of the current techniques and the critical issues that need to be addressed for future breakthroughs. **New and Noteworthy:** This review presents recent advancements in ADPKD organoid construction, particularly iPSC-derived models, offering new insights into disease mechanisms and drug discovery. It focuses on challenges such as limited vascularization and maturity, proposing potential solutions through emerging technologies. The ongoing optimization of ADPKD organoid models is expected to enhance understanding of the disease and drive breakthroughs in disease mechanisms and targeted therapy development.

## 1. Introduction

Polycystic kidney disease (PKD) is the most prevalent hereditary kidney disease; it is mainly divided into ADPKD and autosomal recessive polycystic kidney disease (ARPKD). Among them, ADPKD is the most common monogenic hereditary kidney disorder, with a global prevalence ranging from 1 in 1000 to 1 in 2500, affecting around 12 million people. It is the fourth leading cause of end-stage renal disease (ESRD) [1]. ADPKD is characterized by the formation of numerous fluid-filled cysts within the kidneys, leading to kidney enlargement and impaired function [2]. While most patients remain asymptomatic until adulthood, the disease ultimately progresses to renal failure [3]. The clinical manifestations of ADPKD include hypertension, flank pain, hematuria, urinary tract infections, and gradual decline in renal function. Extrarenal complications are also common and include liver cysts, pancreatic cysts, cardiac valve abnormalities, and intracranial aneurysms [4,5,6,7,8].

In ADPKDs, cyst formation arises from the excessive proliferation of clonal tubular epithelial cells, resulting in cysts that can reach several centimeters in diameter. These cysts can originate in any nephron segment, including the proximal and distal tubules and collecting ducts [9,10]. Structurally, they consist of an epithelial monolayer adhered to an abnormally thickened basement membrane [11]. The process of cyst formation in ADPKD is extremely complex, involving key steps such as the deletion of polycystin, apical–basolateral structural alterations, disruption of cell–matrix interactions, proliferation of cystic cells, and the secretion and accumulation of intracystic fluid [12,13,14,15]. The complexity of these processes makes it difficult to identify specific disease mechanisms in clinical and laboratory settings.

Current research indicates that the primary cause of ADPKD is mutations in the *PKD1* or *PKD2* genes. A rare third genetic locus, *GANAB*, was also identified in 2016 [16]. The *PKD1* and *PKD2* genes encode polycystin-1 (PC1) and polycystin-2 (PC2), respectively, with PC2 playing a critical role in the cilia of renal tubular epithelial cells, participating in calcium signaling and cell proliferation regulation [17]. Over 85% of ADPKD cases are attributed to mutations in the *PKD1* gene encoding PC1, which drives the progression of the disease. The “second hit” hypothesis suggests that a second somatic mutation must occur in addition to germline mutations, leading to the loss of function of PC1 or PC2, which triggers cyst formation [18]. Furthermore, studies have shown that signaling pathways such as mTOR, JAK-STAT, AMPK, and Wnt also play key roles in cyst formation and growth [19,20,21,22].

Currently, ADPKD treatment strategies primarily focus on symptom management and slowing disease progression. Control of hypertension is critical in slowing the deterioration of kidney function, with the use of angiotensin-converting enzyme inhibitors (ACEIs) or angiotensin receptor blockers (ARBs) being common treatment options [23]. However, specific therapies targeting cyst growth remain limited. Tolvaptan, a selective vasopressin V2 receptor antagonist, has been shown to slow kidney volume expansion and decline in renal function, but a limited patient population, liver toxicity, and high costs constrain its use [24,25]. For patients with ESRD, dialysis and kidney transplantation remain the only treatment options [26]. Therefore, the development of novel therapeutic strategies continues to be a key focus in ADPKD research.

Given the complexity of the pathological mechanisms of ADPKD, traditional two-dimensional (2D) cellular models fall short of accurately replicating the detailed physiological processes involved. As a result, the recently developed three-dimensional (3D) organoid model has emerged as a more effective in vitro research tool. These models enable a deeper understanding of the morphological changes in polycystic kidney tissues, underlying disease mechanisms, and the development of therapeutic interventions.

Organoids are cellular constructs grown in a specialized 3D environment in vitro. These constructs undergo self-organization and differentiate into functional cell types through cell sorting and spatially restricted lineage commitment, thereby partially mimicking the functions and structures of real organs in their natural state [27,28]. Due to their specific differentiation potential, pluripotent stem cells are often used as precursor cells for organoids, including embryonic stem cells (ESCs), induced pluripotent stem cells (iPSCs), and adult stem cells (ASCs) [28,29]. They accomplish the internal organization of cells by mediating various signaling pathways through intrinsic cellular components or extrinsic matrix. Self-organization refers to the ability of a cellular system to undergo spatial rearrangement through system-autonomous mechanisms starting from a disordered structure under a uniform signaling environment, which can be classified into self-configuration and morphogenetic rearrangement according to the process [30]. Self-configuration refers to the pattern of cellular differentiation that develops in an initially homogeneous system through autonomous mechanisms within the system and intercellular communication. Morphogenetic rearrangement involves sorting different cell types and higher-level reorganization of the system architecture.

Organoid models closely replicate the natural progression of diseases and can be used to study tissue development, organogenesis, and stem cell behavior in vitro. They hold significant potential for disease modeling and predicting personalized clinical outcomes [31].

This review aims to provide a comprehensive summary of the current advancements and limitations of ADPKD organoid research, highlighting the challenges and opportunities for developing more sophisticated organoid models. By leveraging emerging technologies, this review seeks to inspire future innovations and breakthroughs in the field.

## 2. Molecular and Cellular Mechanisms of ADPKD: Recapitulation in Organoid Models

The first evidence of the kidney’s self-organizing ability emerged from chick embryonic kidney aggregation experiments in the 1960s. These experiments demonstrated that re-aggregated cells from developing chick kidneys could recapitulate nearly complete kidney development [32]. Since 2015, research teams led by Taguchi, Morizane, Freedman, and Takasato have published four distinct protocols for the directed differentiation of kidney organoids [33,34,35,36]. These protocols use combinations of growth factors to mimic endogenous signaling, driving the differentiation process and yielding organoids containing structures such as podocytes, proximal tubules (PT), and distal tubules (DT), thereby laying a solid foundation for ADPKD modeling. Simultaneously, advancements in CRISPR/Cas9 technology have enabled the precise introduction of specific mutations into iPSCs, significantly advancing research into monogenic diseases, including ADPKD [35,37,38].

In the development of ADPKD organoid models, researchers typically aim to first recapitulate the pathological features of ADPKD before validating their consistency with the actual disease mechanisms. Site-specific gene editing techniques, such as CRISPR/Cas9, are commonly employed to generate homozygous or heterozygous mutations in *PKD1* or *PKD2*, thereby modeling the genetic basis of cyst formation [35,39]. However, emerging evidence suggests that genetic modification alone may not be sufficient to fully replicate cystogenesis in vitro. This observation aligns with the widely accepted “second hit” hypothesis, which posits that cyst development requires not only an inherited germline mutation but also a secondary somatic insult, such as environmental stress or additional genetic alterations.

A key experimental validation of this hypothesis is the use of forskolin, a well-established activator of adenylate cyclase (AC), which has been widely applied to promote cystic expansion in epithelial tissues. In ADPKD organoid models, forskolin treatment successfully induces the formation of cyst-like structures, reinforcing its role in driving cystogenesis. Similarly, organoids derived from ADPKD-patient-derived iPSCs require forskolin stimulation to exhibit cyst formation, further supporting the dependence of cyst development on cAMP signaling dysregulation [40,41]. Additionally, it has been hypothesized that disrupted intracellular calcium signaling in ADPKD cells contributes to cystogenesis by inhibiting phosphodiesterase 1/3 (PDE1/3)-mediated cAMP degradation while enhancing AC6-driven cAMP synthesis. Consistent with this hypothesis, nifedipine, a potent L-type calcium channel blocker, has been shown to induce cyst formation in ADPKD kidney organoids while simultaneously elevating intracellular cAMP levels [42]. These findings suggest that ADPKD organoid models can effectively recapitulate the pathological interplay between PC1/PC2-mediated cAMP metabolism and calcium homeostasis during cystogenesis.

Beyond chemical induction, environmental factors are also considered potential triggers of the “second hit”. Injury-induced responses have been proposed as contributors to cyst formation [43,44]. However, Freedman et al. demonstrated that mild injury alone is insufficient to drive cystogenesis in organoids, whereas removal of the extracellular matrix (ECM) appears to play a crucial role [45]. Their study revealed that ECM depletion led to enhanced cell proliferation and migration, potentially creating favorable conditions for cyst formation. These findings suggest that injury responses must act in conjunction with ECM signaling alterations to induce cyst formation in vitro.

Moreover, ADPKD organoid models have successfully recapitulated additional key pathological hallmarks of the disease. For instance, aberrant activation of the renin–angiotensin–aldosterone system (RAAS), a well-documented feature in ADPKD patients, has also been observed in organoids. Specifically, ectopic renin expression was detected in the cystic epithelium and the amount of renin secreted into the culture medium was significantly elevated compared to gene-corrected organoids [42,46]. This successful replication of RAAS dysregulation further reinforces the physiological relevance of ADPKD organoid models in capturing the key molecular mechanisms underlying the disease.

## 3. ADPKD Organoids: Based on iPSCs

Organoids generated from iPSCs represent one of the most versatile and patient-specific models among organoid platforms, effectively mimicking disease states under controlled physiological conditions. Several laboratories have successfully established PKD organoid models using iPSCs, and these studies represent the latest progress in the field of ADPKD organoids.

Roberta Facioli et al. constructed a renal tubule organoid model from erythroid progenitor cells (EP cells) derived from ADPKD patients (Figure 1) [47]. Blood samples were collected from two ADPKD patients and one healthy control, and the EP cells were isolated and expanded. These cells were transfected using episomal vectors from the Epi5 Episomal iPSCs Reprogramming Kit (Thermo Fisher, Waltham, MA, USA) and the Lonza Nucleofector Kit (Lonza, Alpharetta, GA, USA), containing five reprogramming factors, to generate iPSCs. Chromosomal karyotyping confirmed the integrity of the iPSCs, while molecular biology techniques and immunofluorescence verified their pluripotency. Continuous passage enabled these cells to express reprogramming factors successfully. iPSCs were subsequently differentiated into renal organoid structures with proximal tubule characteristics using ROCK signaling inhibitors. Forskolin, a known activator of chloride channels such as CFTR, was added to stimulate cyst formation. This model, based on iPSC reprogramming technology and using easily isolated erythroid progenitor cells, created organoids with a 3D structure similar to renal progenitor cells, offering a convenient and effective platform for studying ADPKD and exploring the molecular and cellular mechanisms of cyst formation.

In another study, Xu Yaoxian et al. identified that CD24^+^ cells are widely distributed in the adult kidney and particularly expressed in the proximal tubules (Figure 1) [48]. Compared to traditional PT cells expressing CD13^+^, CD24^+^ cells exhibit unique metabolic characteristics such as oxygen metabolism, low glycolysis, and specific gene regulatory patterns. A four-stage culture protocol was developed by treating cells with Wnt3a/RSPO1 and stimulating them with epidermal growth factor (EGF)/fibroblast growth factor 2 (FGF2), successfully inducing CD24^+^ cells to generate more tubular epithelial structures known as tubuloids. CRISPR-Cas9 technology was employed to knock out the *PKD1* or *PKD2* genes in these tubuloids, followed by drug treatments to induce cyst formation. Single-cell RNA sequencing revealed that these tubuloids predominantly originated from proximal nephron units, with CD24^+^ cells forming the primary tubular epithelium. This model shared high transcriptomic similarity with tissue samples from ADPKD patients. Immunostaining analysis revealed that renal tubules derived from CD24^+^ cells demonstrated critical features, including apical–basal polarity, cilia presence, well-defined brush borders, and active p-glycoprotein (P-gp) transport activity. These findings highlight that renal tubules comprise various cell types with distinct transcriptomic characteristics. This cellular diversity enables renal tubules to effectively complement the kidney organoids derived from human pluripotent stem cells (hPSCs), providing a robust alternative model for investigating the pathophysiological mechanisms of specific human kidney diseases [49].

Tatsuya Shimizu et al. developed a novel ADPKD model using iPSCs (Figure 1) [40]. CRISPR-Cas9 and the CRONUS system were applied to genetically modify wild-type iPSCs, introducing homozygous or heterozygous *PKD1* mutations that were confirmed via Sanger sequencing. These modified iPSCs were then differentiated into nephron progenitor cells, which formed kidney organoids under air–liquid interface culture conditions after 10 days. Forskolin, a cAMP signaling activator, was added to promote cystogenesis. Additionally, patient-derived iPSC lines carrying heterozygous nonsense or missense *PKD1* mutations underwent similar differentiation and cystogenesis experiments. Immunofluorescence analysis revealed that patient-derived organoids closely mirrored the histological characteristics of *PKD1*-mutated organoids, showcasing the capability of iPSC-derived models to faithfully replicate ADPKD-associated cystogenesis.

In conclusion, iPSC-derived organoids offer significant promise for modeling ADPKD and elucidating the molecular and cellular mechanisms for cyst formation. These models effectively replicate disease-relevant pathophysiological conditions, providing a robust platform for studying ADPKD. Also, iPSC-derived organoids hold considerable potential for advancing drug screening and facilitating the development of gene therapies, paving the way for novel therapeutic approaches to treating ADPKD.

## 4. ADPKD Organoid Models: Compared with Classical Models

### 4.1. In Vitro Cell Models

Among the existing in vitro models for ADPKD, the most commonly used immortalized animal kidney cell lines are Madin–Darby canine kidney (MDCK) cells and porcine kidney proximal tubule (LLC-PK1) cells (Table 1). McAteer et al. demonstrated that MDCK cells could suspend and proliferate to form epithelial cysts when seeded into moderately hydrated collagen gels [50,51]. Further studies revealed that cyst enlargement in this system is influenced by the cyst density within the gel, the medium composition, and the mechanical properties of the collagen substrate [52]. Additionally, the transparent nature of collagen gels facilitates direct observation of cyst formation using optical microscopy. Building on this, Grantham and colleagues found that adding cAMP agonists could promote cyst formation in MDCK cells, thereby providing a valuable platform for studying fluid secretion mechanisms in cystogenesis [53]. By mimicking solute transport processes, this model has proven instrumental in elucidating the dynamics of cyst expansion, offering a straightforward and effective research tool.

LLC-PK1 cells, derived from porcine kidney proximal tubules, have similar physiological and structural characteristics to human proximal tubule cells and express the key ADPKD-related proteins PC1 and PC2. When cultured in suspension, LLC-PK1 cells spontaneously aggregate to form freely floating hollow spheroids or cystic structures, recapitulating the key characteristics of cystogenesis in ADPKD [54,55].

Despite the practical advantages of MDCK and LLC-PK1 models—including ease of manipulation and high reproducibility—their application is constrained by species-specific differences, which limit their translational relevance for modeling human disease. These animal-derived cells tend to form spherical cyst-like structures in vitro, which do not fully reflect the structural characteristics of pathological cysts in ADPKD [56]. In contrast, patient-derived cell lines offer greater clinical relevance. Mahmoud et al. established two human ADPKD cell lines, WT9-7 and WT9-12, by isolating epithelial cells from 30 cysts obtained from 11 ADPKD patients and immortalizing them using adenovirus and SV40 transformation techniques (Table 1) [57]. WT9-7 cells, derived from proximal tubules, harbor heterozygous *PKD1* truncation mutations, whereas WT9-12 cells, which exhibit characteristics of both proximal and distal tubules, contain homozygous *PKD1* mutations. These cell lines exhibit high genetic stability and growth consistency, making them valuable tools for studying the region-specific pathological mechanisms of ADPKD in different segments of the nephron.
biomedicines-13-00523-t001_Table 1Table 1Summary of in vitro cell models.ModelSpeciesTubule OriginMutation MechanismReferenceMadin–Darby Canine Kidney Cell (MDCK)CanineDistal tubuleNone[50,51]Porcine Kidney Proximal Tubule Cells (LLC-PK1)PigProximal tubuleNone[54,55]WT9-7HumanProximal tubuleA truncating mutation in one allele of *PKD1*, resulting in partial loss of polycystin-1 function.[57]WT9-12HumanProximal tubule and distal tubuleA homozygous nonsense mutation in *PKD1*, causing complete loss of polycystin-1 function.[57]


### 4.2. In Vivo Animal Models

The intricate genetic landscape and unique 3D pathology of ADPKD pose significant challenges for its modeling, as no single cellular model can fully recapitulate its complex disease features. To address this, researchers have developed a diverse array of animal models spanning the evolutionary spectrum, from simple invertebrates (e.g., Drosophila), through lower vertebrates (e.g., African clawed toad and zebrafish), to higher mammals (e.g., mice, rats, cats, pigs, horses, and monkeys) [58,59,60,61,62]. Among these, rodent models have emerged as particularly valuable tools for studying renal cyst formation and disease progression owing to their well-characterized genetics and adaptability to various experimental manipulations [63,64].

Mouse models, in particular, have become the cornerstone of ADPKD research. Through sophisticated gene editing techniques and natural mutation screening, several key mouse models have been established to address different aspects of the disease (Table 2). Since homozygous germline deletion of *Pkd1* or *Pkd2* in mice leads to embryonic lethality, conditional or kidney-specific knockout models are more suitable for studying the pathogenesis of ADPKD [65]. For instance, point mutation models, such as *Pkd1*^RC/RC^ mice that carry the R3277C mutation in the *Pkd1* gene, effectively mimic the chronic progression of human ADPKD, characterized by delayed cyst formation. These models are particularly useful for investigating the molecular mechanisms underlying cystogenesis and disease progression [66,67]. Conditional knockout models employing the Cre-loxP system allow for tissue- or time-specific gene inactivation. A notable example is the *Pkd1* flox/− Ksp-Cre/− mouse, where the Ksp-Cre promoter drives the specific knockdown of *Pkd1* in renal tubular epithelial cells during embryonic development, resulting in rapid cyst formation postnatally. This model has become a classic tool for studying the early pathogenesis of ADPKD [68]. Additionally, the fusion of Cre recombinase with the estrogen receptor (ERT2) to create CreERT2 enables temporal control of gene knockdown through tamoxifen administration, facilitating the study of the temporal relationship between gene inactivation and cyst formation [69]. Another intriguing category includes natural mutation models such as the *pcy* mouse, which spontaneously develops renal cysts due to mutations in the *Bicc1* gene. This model provides unique insights into the role of non-Pkd genes in cystogenesis [70].

Rat models complement mouse models by addressing some of their limitations, particularly the short lifespan of mice. The *Han: SPRD* rat model, the first naturally occurring rat model of PKD, is widely used in ADPKD research (Table 2) [71]. This model exhibits several features reminiscent of human ADPKD, including autosomal dominant inheritance, renal cyst formation, and extrarenal manifestations. The model originated in 1986 when researchers identified male rats with hereditary bilateral PKD in the *Han: SPRD* breeding colony [72]. Through 18 generations of inbreeding, Dr. Gretz established the *PKD/Mhm* (University of Heidelberg, Mannheim, Germany) rat strain [73,74,75]. While the *Han: SPRD* rat model effectively simulates key pathological features of ADPKD, it has notable limitations, including pronounced sex differences (with males exhibiting more severe disease than females), cyst formation primarily in the proximal tubules, and the absence of multisystemic pathology [72]. Importantly, the genetic basis of this model differs from human ADPKD, as it involves mutations in the *PKDr1* gene, which is homologous to *PKD1* [73].

Traditional in vitro cellular models and in vivo animal models have been indispensable in advancing our understanding of ADPKD. Their diversity and flexibility have provided a robust foundation for dissecting disease mechanisms and exploring therapeutic avenues. However, the advent of iPSC-derived organoid models has introduced a new dimension to ADPKD research. These models, derived from patient-specific iPSCs, offer superior capabilities in mimicking the genetic complexity and three-dimensional pathological architecture of ADPKD. Organoid models excel in replicating the developmental processes and pathological features of human renal units, particularly in the context of cyst formation and development [35].

Compared to conventional cell lines such as MDCK and LLC-PK1, organoid models can more accurately reflect the diverse genetic backgrounds of ADPKD through advanced gene editing technologies, thereby enhancing their clinical relevance. Moreover, organoid models can rapidly generate cystic structures under 3D culture conditions, making them ideal for high-throughput drug screening. This capability not only mitigates the variability inherent in animal models but also aligns with the ethical principles of the 3Rs (Replacement, Reduction, and Refinement), promoting more sustainable research practices.

Despite their advantages, organoid models are not without limitations. For instance, they lack a fully developed immune system and vascular network, which restricts their ability to fully replicate the complex cell–cell interactions and dynamic microenvironment present in vivo [28]. To overcome these challenges, researchers have begun integrating organoid models with in vivo animal models. For example, signaling pathway alterations identified in animal models can be validated using organoids, followed by screening for potential therapeutic targets. This integrative approach not only addresses the limitations of individual models but also opens new avenues for a more comprehensive understanding of ADPKD pathogenesis [76].

## 5. ADPKD Organoid Application: From Mechanism to Treatment

### 5.1. Exploring the Pathogenesis of Disease

As an in vitro model, ADPKD organoids effectively replicate the onset and progression of the disease, providing a robust platform for studying cystogenesis. A range of compounds, including EGF, blebbistatin, cisplatin, and antidiuretic hormone (AVP), have been tested on patient-derived iPSC-induced organoids to access their potential to induce cyst formation in ADPKD models [40]. Chemical screening using patient-derived renal organoids has further enabled the identification of the key pathways involved in cystogenesis, revealing the critical mechanisms underlying ADPKD progression.

In another experiment, *PKD1^−/−^* organoids embedded in collagen droplets (~4 mm) displayed significantly impaired collagen contraction compared to controls, underscoring the essential role of PC1 in maintaining renal structure [45]. Recent research has also identified a possible myosin-heavy-chain-like calmodulin-binding structural domain at the carboxyl terminus of PC1, suggesting its involvement in myosin regulation [77]. Myosin is known to be involved in intercellular adhesion junctions and tight junctions, particularly in formation of purse-string-like contractile rings [78,79]. Consequently, PC1 may activate actin filaments in tubular epithelial cells, enhancing the intercellular contraction of renal tubules and preventing them from deforming into cysts [31].

In a notable breakthrough, Freedman et al. developed a microfluidic system for culturing ADPKD organoids in a breathable flow chamber. Organoids carrying *PKD1* mutations exhibited rapid cyst enlargement under flow conditions [80]. In contrast, both control organoids and those cultured under static conditions displayed minimal swelling. Furthermore, the study demonstrated that, aside from fluid flow, factors such as volume and solute concentration act as positive regulators of cyst expansion. These findings offer new perspectives into the pathological mechanisms of ADPKD and highlight potential therapeutic targets for future interventions.

### 5.2. Bridging the Gap from Model to Therapy

Drug development is a challenging and resource-intensive process, with a high failure rate even for the most promising candidate molecules that progress to preclinical studies or clinical trials [81]. Therefore, early-stage screening and validation using organoid models can significantly enhance the success rate of drug development by providing a more physiologically relevant platform, while also reducing the associated costs and time investment.

In 2022, Tran et al. developed an innovative platform to mimic ADPKD by inducing mutations in the *PKD1* and *PKD2* genes in hPSCs [82]. After approximately two weeks of culture, these mutant organoids exhibited cystic structures. Using a mass production strategy, the researchers generated methylcellulose-embedded organoids in 96-well plates for high-throughput screening (HTS). Their approach successfully identified several known cyst inhibitors, including a novel compound called quinazoline, demonstrating the platform’s effectiveness for drug discovery. Similarly, Biao Huang et al. successfully generated 3D renal metazoan progenitor cells (NPCs) by knocking down *PKD1* and *PKD2* using the CRISPR-Cas9 system [83]. These NPC aggregates were subsequently subjected to a cyst-inhibition drug screen using a small molecule library containing 148 compounds, resulting in the identification of 14 compounds with cyst-inhibiting properties. Among these, metformin showed dose-dependent inhibition of cyst formation.

Furthermore, kidney-like organs generated from patient-derived iPSCs present a promising tool for screening personalized therapeutic regimens. In another study, researchers utilized patient iPSC-derived kidney-like organs to evaluate their responses to known inhibitors, comparing the results to the responses observed in patient kidneys [40]. The drugs tested included tolvaptan, a selective V2R antagonist, the cystic fibrosis transmembrane conductance regulator (CFTR) inhibitor 172, and everolimus, a mammalian target of rapamycin (mTOR) inhibitor. Among these, tolvaptan showed limited efficacy. The lack of response to tolvaptan is likely attributable to the absence of Arginine Vasopressin Receptor 2(AVPR2) receptors in these organoids [83]. This finding underscores the importance of considering mechanistic differences when using organoid models for drug screening [82]. In conclusion, while organoid models are valuable tools for drug screening and validation, continuous optimization and refinement are essential to enhance their accuracy and applicability in drug development, disease research, and personalized medicine. The use of organoids for automated HTS of large-scale compounds or genes offers a scale unattainable with mammalian model organisms, making this one of the most compelling potential applications for organoid applications [84].

In addition to drug screening, gene therapy for correcting genetic mutations has emerged as a cutting-edge strategy for treating ADPKD. The majority of *PKD1* mutations in ADPKD are single-nucleotide variants, making CRISPR-Cas9-based adenine base editing (ABE) technology particularly attractive due to its minimal off-target effects [85]. To assess the feasibility of a gene-editing approach targeting nonsense mutations for PKD therapy, researchers first developed a homogenous hPSC (A/A) organoid model carrying the common PKD2-R186X nonsense mutation [86]. The PKD2-R186X mutation lies within the DNA editing window, allowing for precise correction via adenine base editing without inducing bystander effects. Using plasmid vectors for delivery, researchers transfected hPSCs and employed fluorescence-activated cell sorting (FACS) to isolate successfully transfected cells. In heterozygous gene-corrected (A/G) clones, PC2 protein levels were restored to approximately 50% of those observed in non-mutated cells, representing a significant improvement compared to the unedited PKD2-R186X homozygous mutants (A/A). Furthermore, correcting the PKD2-R186X mutation reduced cyst formation rates from over 70% in the unedited homozygous mutant organoids to approximately 10% in the heterozygous gene-corrected organoids, with a marked reduction in cyst cross-sectional area.

In a separate study, researchers similarly employed ABE to repair a clinically identified pathogenic *PKD1* point mutation, c.8311 (G > A) [87]. For delivery, adeno-associated virus (AAV) was chosen as an ideal vector due to its low immunogenicity, good safety profile, and prolonged in vivo expression. However, the limited packaging capacity of AAV precluded the direct delivery of the complete ABE system. To circumvent this limitation, researchers utilized intein-mediated protein trans-splicing to split ABE into two smaller functional units that were then delivered via a dual-AAV system [88,89]. Subsequent experiments evaluated the infection efficiency of different AAV serotypes (including AAV2, AAV6, and AAV9), revealing that AAV6 exhibited the highest transfection efficiency, significantly enhancing the editing outcomes.

The commonly used vectors for gene delivery include lentivirus, adenovirus, AAV, and plasmid DNA, with selection criteria depending on factors such as gene size, duration of expression, and safety profile [90,91]. Lentiviral vectors are suitable for long-term expression and adenoviruses offer strong infectivity but higher immunogenicity, while AAV is favored for its low immunogenicity and sustained expression [92]. Although gene therapy for ADPKD using organoid models remains in the exploratory phase, with no specific vector yet widely adopted, ongoing technological advancements are expected to yield more optimized delivery systems in the near future [93].

### 5.3. Ethical Considerations: Unignorable Issues in ADPKD Organoids

In the preceding sections, we have explored the generation of organoid models from iPSCs and their latest advancements in elucidating disease mechanisms and developing therapeutic strategies. It is undeniable that iPSC-derived ADPKD organoids demonstrate significant potential in disease modeling, personalized medicine, and drug screening. However, as these technologies continue to evolve, their associated ethical concerns have become increasingly prominent [94]. Currently, some ADPKD organoids are derived from patient-specific iPSCs, necessitating rigorous ethical review prior to obtaining somatic cells from patients. This ensures a reasonable balance between the potential risks and benefits while fully considering the direct benefits that donors may receive [95,96]. Moreover, since iPSCs are derived from the somatic cells of identifiable individuals, researchers must disclose the intended research uses of the donor-derived iPSCs and obtain fully informed consent. This process is crucial for establishing cell line repositories, sharing research resources, and ensuring the ethical utilization of donated biological samples in the future [97,98,99]. Notably, the establishment and use of disease-specific iPSC lines have become a critical component of drug development. Three-dimensional organoid models can more accurately mimic the disease microenvironment, significantly enhancing the efficiency of drug screening, reducing research costs, and improving the reliability of drug safety assessments [100].

Furthermore, the application of gene editing technologies in iPSC-derived organoids has sparked new ethical debates. The use of gene editing typically involves the acquisition of patient cells, genetic modifications, and the storage and use of associated data, raising concerns about the privacy protection of patient biological data and the need for ethical oversight. Although gene editing technologies have not yet been widely applied in clinical treatments for ADPKD patients, the continuous optimization of CRISPR-Cas9 and its derivative technologies has brought to the forefront the question of how to regulate the application of gene editing. Ensuring that these technologies do not induce off-target effects or other unforeseen adverse consequences has become a focal point for both the scientific and ethical communities.

In conclusion, while iPSC-derived organoids and gene editing technologies hold immense promise for ADPKD research and treatment, the advancement of these technologies must be accompanied by the development of robust ethical frameworks. This will ensure that their applications remain compliant with societal and legal standards, thereby mitigating potential ethical and legal challenges.

## 6. Limitations of Autosomal Dominant Polycystic Kidney Disease Organoids

Despite the significant advancements in ADPKD organoids for simulating disease processes and studying the underlying mechanisms, several challenges and limitations persist. Addressing these issues is essential to enhance the utility of organoid models in disease research and drug screening.

### 6.1. Simulation of the Real Environment

Current ADPKD organoid models are primarily generated by inducing mutations in the *PKD1* and *PKD2* genes in iPSCs. While these models successfully replicate many of the disease mechanisms observed in patients, they predominantly involve pure gene deletions. In contrast, ADPKD patients typically carry heterozygous mutations, suggesting that these organoids may not fully capture the realistic genetic profile of the disease. This discrepancy raises concerns about their ability to accurately reflect the complex genetic landscape of ADPKD [101].

Furthermore, accurately modeling the “second hit” hypothesis—a central feature in ADPKD pathogenesis—remains a significant challenge. Although some studies have attempted to induce this process through forskolin treatment, it remains unclear whether cystogenesis necessarily requires a “second hit” beyond *PKD1* and *PKD2* mutations or what constitutes this second event [102,103]. Clarifying this mechanism is critical for advancing our understanding of ADPKD.

Another limitation lies in the distribution of cysts within the organoid models. In ADPKD organoid models, cysts formation predominantly occurs in the proximal tubules [41], whereas in ADPKD patients, cysts are more frequently observed in the distal tubules or collecting ducts [104]. This disparity in cyst localization further impacts the translational relevance of these models and limits their ability to accurately simulate the full spectrum of ADPKD pathology.

These limitations underscore the need for the continued optimization of ADPKD organoid models to better capture the genetic and structural complexities of the disease. Addressing these issues is crucial to improving our understanding of disease mechanisms and advancing the development of effective therapeutic strategies. Future research should focus on enhancing the genetic fidelity and structural accuracy of ADPKD organoids, ensuring their reliability as comprehensive disease models (Figure 2).

### 6.2. Vascularization of Organoids

Despite their relatively small size, the kidneys receive a substantial blood supply, accounting for approximately 20–25% of the body’s total cardiac output [105]. This extensive perfusion is crucial for their functions in filtration, electrolyte balance, and acid–base homeostasis. Consequently, achieving effective vascularization in in vitro-cultured ADPKD organoids remains a significant technical challenge, limiting their long-term viability and functionality.

Adequate vascularization is essential for the survival and proper function of renal organoids [106]. One study demonstrated that disrupting the embryonic heartbeat in zebrafish using 2,3-butanedione monoxime (BDM) inhibited the fusion of glomerular precursor cells, thereby preventing glomerular formation [107]. Restoration of the heartbeat upon BDM removal resumed glomerulus formation. Further experiments revealed that local laser-induced obstruction of blood vessels also impeded glomerular formation, even when the heartbeat was present, underscoring the critical role of blood flow in renal vascularization.

Interestingly, stem-cell-derived renal organoids contain a small number of residual vascular endothelial cells. However, in in vitro conditions, these endothelial cells fail to form vascular networks and are progressively lost over time in culture [108]. Even the addition of vascular endothelial growth factor (VEGF) during differentiation, which increases endothelial cell (EC) numbers by approximately tenfold, is insufficient for forming functional vascular networks within the organoid [31]. In contrast, human iPSC-derived glomeruli exhibit effective vascularization when nephron progenitor cell (NPC)-type organoids (lacking ureteric bud, UB) are transplanted under the kidney capsule of immunodeficient mice [108]. Characterization of these organoids indicated that most of the endothelial cells in the glomeruli originated from the host rather than the transplanted organoid [109,110]. Furthermore, human iPSC-derived vascular endothelial cells can integrate with host endothelial cells to varying degrees post-transplant [111,112].

These findings raise new questions about the role of host endothelial cells in organoid vascularization. Further studies are required to clarify the contribution of both donor- and host-derived endothelial cells in the vascularization of transplanted organoids. Specifically, it remains to be determined whether inducing endothelial cells from human iPSCs before transplantation is necessary and whether it is necessary to induce endothelial cells from human iPSCs prior to transplantation and whether these cells should be integrated into the organoid before transplantation. If exogenous endothelial cells are required, it will be crucial to precisely characterize the specific subpopulation of these cells.

Microfluidic technologies offer a promising approach for enhancing organoid vascularization. These systems provide dynamic fluidic shear stress (FSS), a critical environmental signal that stimulates endothelial progenitor cell (EPC) proliferation and vascular network formation. Studies have shown that the application of FSS via microfluidic devices can significantly improve the vascularization and maturation of organoids [113]. Classical organs exposed to higher FSS conditions exhibited enhanced peripheral expression of the vascular markers, such as melanoma cell adhesion molecule (MCAM) and platelet endothelial cell adhesion molecule (PECAM), along with a tenfold increase in the density of vascular connections (number of branch points per unit area) and the average vessel length (distance between connections) [114]. Therefore, further optimization of the microfluidic device, including modulation of the fluid shear and pressure, could more accurately replicate the complex blood flow environment and physiological conditions in vivo, thereby promoting the formation of more physiologically relevant vascular networks within organoids (Figure 2).

### 6.3. Irreducibility and Cellular Heterogeneity

Since the initial report of human pluripotent stem-cell-derived kidney-like organs, a growing body of literature has emerged, presenting various methods for inducing kidney-like structures. However, significant heterogeneity exists among the organoids generated by different laboratories. Variations in induction reagents, culture conditions, and methodologies inevitably lead to differences in organoid composition and characteristics. This issue is especially pronounced when using iPSCs derived from human patients, as the resulting organoids exhibit substantial variability in cell lineage differentiation, complicating the analysis of disease-specific effects, such as those observed in ADPKD [45]. Notably, even when using a standardized protocol with the same iPSC line, considerable variability can still occur between experiments [104].

To enhance reproducibility in organoid generation, it is crucial first to identify the root causes of this variability. Both the Moriza’s and Taguch’s protocols suggest that variations in endogenous bone morphogenetic protein (BMP) signaling may contribute to clonal variability, though direct evidence remains limited [115,116]. Another study utilizing the RNA sequencing (RNA-seq) of various renal organoids indicated that alterations in nephron patterning and cellular component ratios are major sources of variability [117].

Standardizing the culture conditions is essential for enhancing experimental reproducibility and ensuring cellular consistency. This can be achieved by strictly controlling variables such as culture temperature, humidity, and CO_2_ levels to maintain environmental stability. Additionally, developing and using commercially available, validated culture media along with standardized culture vessels and equipment can further reduce variability across experiments. Ensuring consistency in cell source and handling is also vital. Employing the same batch of cell lines or biomaterials, coupled with uniform cell processing methods, helps to minimize batch-to-batch variation [118]. For monogenic diseases such as PKD, the use of CRISPR/Cas9 technology enables the creation of homologous corrected lines in organoids derived from patient iPSCs, which can reduce the batch effects that occurs when comparing with mutant lines [119].

Single-cell cloning offers a powerful approach to reducing initial population heterogeneity by generating organoids from individual cell clones. Techniques such as fluorescence-activated cell sorting (FACS) can facilitate the isolation of individual cells, ensuring that each organoid is derived from a homogeneous cell population [120].

Establishing and sharing standard operating procedures (SOPs) for organoid culture and data analysis, alongside the application of bioinformatics tools to integrate large-scale organoid data, will help identify the key factors affecting reproducibility and cellular heterogeneity. Research by Phipson et al. (2019), which analyzed the transcriptional and morphological characteristics of organoids derived from the same human iPSCs line across six independent differentiation experiments, identified 10 biomarkers that can accurately predict relative organoid maturation [117].

By addressing these technical challenges, organoid models can be made more reproducible and reliable, facilitating advancements in disease modeling and drug discovery [121] (Figure 2).

## 7. Challenges and Prospects in Autosomal Dominant Polycystic Kidney Organoid Research

### 7.1. Generation of Complex Renal Structures

The human kidney is a highly intricate organ, comprising over 25 distinct cell types and containing approximately 200,000 to 2 million nephrons that are organized within the cortical and medullary regions of the organ [122,123]. Each nephron consists of a glomerulus and a renal tubule. The glomerulus is connected to the systemic vasculature via afferent arterioles, while the renal tubule connects to a central collecting duct system. This duct system collects urine from multiple nephrons and channels it into the renal pelvis, where it is excreted through the ureter [124].

In vivo, the development of this highly complex renal architecture takes approximately 200 days [125]. In contrast, most in vitro organoids are cultivated for only a few weeks, resulting in structures that lack the full complexity and integrity of mature kidneys. Additionally, when generating organoids from iPSCs, single-cell lines are typically used to induce specific structures. This approach has often led to the production of organoids that do not fully replicate the architecture of an intact kidney. This limitation not only affects the overall accuracy of organoid-based research but also compromises the comprehensiveness of drug screening and hinders the translational application of organoid models from laboratory to clinical practice. For instance, if an organoid model lacks AVPR2, potential novel AVPR2 antagonists may be overlooked during drug screening, thereby limiting the identification of effective therapeutic candidates [83].

To address this limitation, some researchers have applied 3D printing technology into organoid development, a technique referred to as “organoid printing” [126]. In this process, dissociated single-cell suspensions are formed into a cell paste without any carrier hydrogel. This paste is then loaded into Hamilton syringes and deposited using an automatically NovoGen MMX extrusion-based 3D cell bioprinter (Organovo, San Diego, CA, USA). This technology enables the rapid, high-throughput production of kidney-like organoids with more structured and organized cell arrangements. Furthermore, 3D bioprinting allows for precise control over biophysical properties, including organoid size, cell number, and spatial configuration.

Furthermore, advancements in imaging technology have significantly contributed to the development of organoid models. These innovations are crucial for deciphering organoid complexity, mapping their morphology, and validating their accuracy relative to their in vivo counterparts. In recent years, quantitative phase imaging (QPI) has gained recognition for its ability to perform label-free imaging of living biological samples [127,128,129]. Holographic tomography (HT), a 3D extension of QPI, enables real-time visualization of dynamic cellular changes within organoids without inducing phototoxicity or photobleaching. Consequently, HT technology has been widely applied in assessing organoid structural development and pharmacological responses [130]. Researchers have employed low-coherence HT for whole-layer scanning and image reconstruction of murine small intestinal organoids (sIOs), elucidating their intricate internal architecture and revealing complex processes such as brush border formation and the accumulation of shed cells within the apical lumen.

With the advancement of research, the continuous development of 3D bioprinting and high-resolution imaging technologies has provided more powerful tools for the reconstruction, observation, and analysis of organoid structure, function, and pathological changes. These cutting-edge technologies offer new opportunities to enhance organoid fidelity, improve drug screening accuracy, and increase the clinical relevance of organoid models, thereby facilitating their application in personalized medicine and translational research (Figure 2).

### 7.2. Disease Staging Simulation

In patients with ADPKD, multiple epithelial-lined cysts develop within the kidneys, leading to a gradual and substantial enlargement of the organs. Most individuals with ADPKD eventually progress to renal failure by their fifth or sixth decade of life [26]. Over the course of this prolonged disease progression, the pathological phenotype of the kidneys deteriorates progressively. However, current organoid-based research predominantly focuses on cyst formation as the primary endpoint, lacking a comprehensive exploration of the broader aspects of cystogenesis and full disease progression.

During the formation of organoids, analyzing their dynamic morphological changes is essential for assessing their maturation and gaining a deeper understanding of disease progression at different stages. However, organoid imaging analysis presents significant challenges, primarily since images are typically acquired in a single focal plane, and even within the same organoid model, considerable variations in size and shape can be observed. While genetic modifications can be employed to induce the expression of fluorescent proteins to facilitate image segmentation and tracking, this approach not only increases experimental complexity and duration but may also alter the intrinsic cellular dynamics of the original sample. Therefore, integrating machine-learning-based image analysis to automatically extract morphological features and dynamically assess organoid development across different growth stages could provide a more accurate evaluation of organoid maturation and progression [131].

Similarly, the field of cancer organoids has successfully employed genetic reprogramming to model different stages of tumorigenesis, enabling researchers to suppress oncogenic phenotypes and investigate the early stages of disease development [132]. This approach could similarly be applied to ADPKD models by introducing stage-specific mutations through CRISPR/Cas9 technology, thereby simulating distinct phases of ADPKD organoid progression. For example, during the construction of ADPKD organoids, CRISPR/Cas9 technology facilitates the precise introduction of mutations in the *PKD1* or *PKD2* genes while also enabling the simulation of the “second hit” phenomenon at multiple time points. This approach involves introducing additional mutations on top of the initial somatic mutation, thereby accelerating cyst formation. Through this process, organoids can progressively recapitulate the key pathological features of ADPKD, including cyst expansion, kidney enlargement, and the gradual decline in renal function.

In recent years, integrating RNA sequencing (RNA-seq) with time-series analysis has provided a novel analytical framework for investigating the dynamic changes in organoids. By leveraging time-series RNA-seq data, researchers can identify and quantify the key genes and pathways associated with disease progression. For instance, a study identified ten genes exhibiting strong linear correlations during differentiation for 7 to 25 days. These data were further utilized to establish a multiple linear regression model for estimating the “age” of organoids, enabling precise tracking of their developmental dynamics [117].

The combination of machine learning-based image analysis, RNA sequencing, and time-series analysis represents a multidimensional approach that offers a valuable framework for establishing stage-specific ADPKD organoid models. This integrated methodology allows researchers to comprehensively explore the phenotypic changes of organoids across different developmental stages. Moreover, in conjunction with CRISPR/Cas9 and other gene-editing technologies, researchers can selectively manipulate key pathways and genes at specific stages of organoid development, further elucidating the molecular alterations in ADPKD organoids. This approach provides deeper insights into disease progression and enhances our understanding of the pathophysiological mechanisms underlying ADPKD (Figure 2).

### 7.3. Organoid Microenvironment Simulation

ECM is crucial in supporting cell culture systems as they transition from 2D to 3D models [28]. In organoid culture, incorporating 3D scaffolds that mimic the in vivo ECM, particularly using natural biomaterials, provides essential mechanical support for cell–cell interactions and recreates the kidney’s ECM environment. Currently, the most commonly used ECM hydrogel in organoid culture is Matrigel, derived from the ECM secreted by the Engelbreth–Holm–Swarm tumor line, which is rich in laminin [133]. However, as an animal-derived ECM, Matrigel presents challenges such as batch-to-batch variability, limited reproducibility, and potential immunogenicity. Synthetic hydrogel matrices have emerged as a promising alternative due to their relatively low cost, high reproducibility, and ease of purification and manipulation. Research suggests that the kidney-specific ECM, composed of collagen, elastin, proteoglycans, glycoproteins, and growth factors, is essential for kidney development and function [134]. Tailoring the ECM composition to the specific requirements of the organoid can significantly improve its structural and functional fidelity. Machine learning offers a powerful solution to the inherent complexities of designing hydrogels for specific applications, including organoid research [131]. By screening large datasets, machine learning can facilitate the analysis of correlations between the biomaterial properties and organoid culture outcomes, providing valuable insights that accelerate the development of customized matrix gels. Studies have shown that the different properties of the ECM can promote distinct differentiation processes in organoids [135]. For instance, softer matrix environments (approximately 10–20 kPa) have been shown to enhance organoid self-organization, cell cohesion, and the expression of mesodermal and kidney-specific genes, leading to more mature structures compared to those grown on stiffer matrices [136,137]. Developing novel ECM materials could provide more precise growth conditions for organoids, further optimizing their physical support and functionality.

Improvements in co-culture systems have also enhanced the realism of cell–cell interactions in organoids. For example, co-culturing fibroblasts and immune cells with cancer organoids has shown that these models can better simulate the tumor microenvironment, offering insights into how these cells influence tumor growth and invasion [138]. Similarly, in ADPKD, abnormal cell proliferation and uncontrolled growth result from dysregulated cellular signaling pathways, where the microenvironment plays a crucial role in disease development and progression. However, the current kidney organoid models remain relatively rudimentary, often lacking essential components necessary for organ function, such as mesenchymal cells, immune cells, and stromal cells. Evidence suggests that co-culturing kidney organoids with endothelial and stromal cells can promote cellular assembly and maturation.

In one study, MDCK were embedded in a gel layer containing mouse embryo-derived Swiss 3T3 or BALB 3T3 fibroblasts, allowing co-culture of MDCK and 3T3 cells. The results demonstrated that suspended MDCK cells formed smooth aggregates and developed central lumens with apical microvilli and junctional complexes. However, communication between the lumens and the surrounding medium was not observed [139]. Similarly, the cytokines and growth factors secreted by stromal cells have been shown to enhance the vascularization and maturation of kidney organoids [140]. For instance, Xinaris et al. developed a 3D chimeric organoid by co-culturing mouse embryonic kidney-derived cells with human amniotic fluid stem cells (hAFSCs) [141]. The expression of glial-cell-line-derived neurotrophic factor (GDNF) by hAFSCs, a key growth factor, promotes the integration of the organoid with the host tissue post-transplantation, leading to the formation of vascularized glomeruli and highly mature tubular structures.

In the context of ADPKD research, co-culture systems not only increase the complexity of organoids but also provide a platform to investigate whether the cytokines produced in these systems could act as initiating factors in the “second hit” hypothesis. This approach provides a promising direction for studying cyst formation mechanisms and developing innovative therapeutic strategies.

Recent advancements in 3D media, co-culture systems, and chemical signaling regulation are driving the simulation of organoid microenvironments toward greater precision and complexity, allowing for more accurate approximations of the actual pathophysiological conditions (Figure 2).

This schematic illustrates the main challenges facing PKD organoid technology, including accurate simulation of real disease conditions, vascularization, and difficulty in reproducing the culture process. The diagram also outlines potential solutions to overcome these technological bottlenecks through 3D printing technology, application of extracellular matrix, and co-culture systems. In addition, PKD organoids may provide an important research platform for more accurate disease staging simulations in the future.

## 8. Conclusions

Over the past five years, iPSC-derived ADPKD organoids have gained considerable attention, leading to the development of various models exhibiting ADPKD phenotypes. These miniature in vitro structures provide novel opportunities to investigate disease pathogenesis, explore the complex mechanisms underlying disease progression, and facilitate drug discovery and testing. In this process, the application of iPSC-derived organoids is accompanied by ethical considerations that cannot be ignored. However, despite these promising advancements, challenges such as limited vascularization and incomplete maturation remain, constraining the full potential of these models. As emerging technologies continue to evolve, we are optimistic that addressing these technical challenges will enable the creation of more sophisticated ADPKD organoids that can faithfully replicate the disease processes in a physiological environment, thereby accelerating breakthroughs in research and therapeutic applications.

## Figures and Tables

**Figure 1 biomedicines-13-00523-f001:**
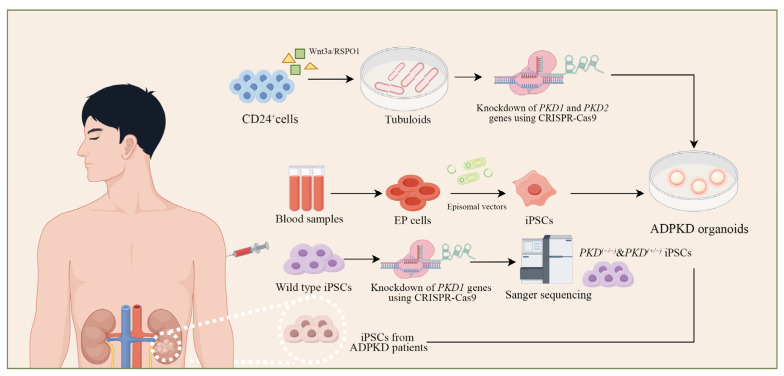
Different protocols to generate ADPKD organoids. Currently, various experimental protocols based on iPSCs have been developed to induce the formation of ADPKD organoids. These protocols include the use of EP cells, CD24^+^ renal PT cells, and patient-derived iPSCs to model the pathogenesis of ADPKD.

**Figure 2 biomedicines-13-00523-f002:**
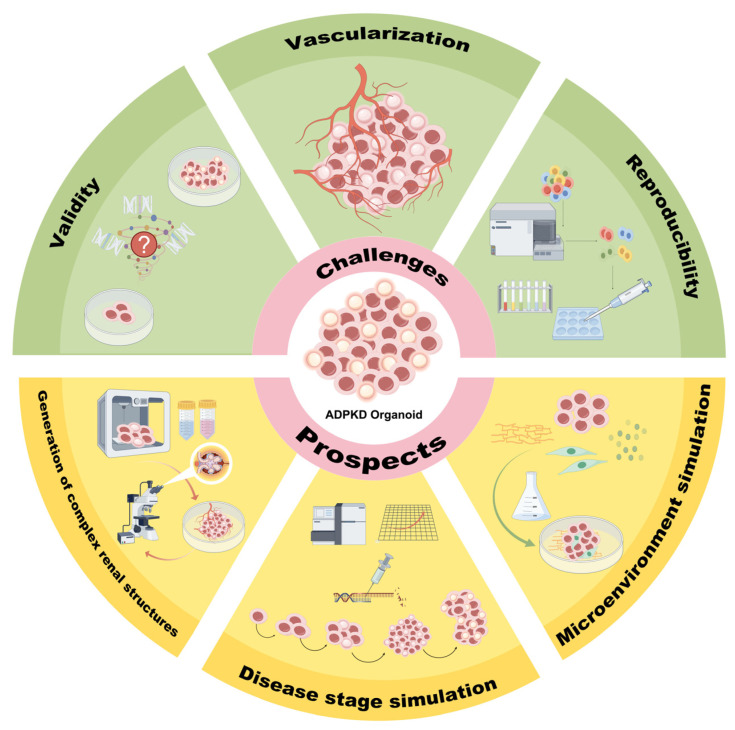
Challenges and prospects of ADPKD organoids.

**Table 2 biomedicines-13-00523-t002:** Summary of in vivo animal models.

Model	Mutation Mechanism	Human Gene	Disease Stage	Survival	Reference
**Mice** *Pkd1* ^RC/RC^	Missense mutation mimicked from human patients (p.R3277C); approximately 40% mature PC1	*PKD1*	Early and late stage	Normal ~18–24 months	[66,67]
*Pkd1* flox/−; Ksp-Cre/−	Deletion of exons 2–4	*PKD1*	Early stage	P14–P17	[68]
*Pkd1* flox/−; Cdh16 −cre/ERT2	Deletion of exons 2–11	*PKD1*	Early and late stage	not described	[69]
*Pyc* mouse	Spontaneous mutation: Intronic mutation in *Bicc1*	*BICC1* (Non-*PKD* gene but involved in cystogenesis)	Late stage	~12–15 months	[70]
**Rat** *Han:SPRD Rats*	Spontaneous mutation: *PKDr1* gene mutation	*PKD1* (Partial mimicry of *PKD1*)	Late stage	~12–18 months	[71,72]

Classification of disease stages: (i) Early stage: Cyst formation or renal dysfunction occurs shortly after birth (<1 month). (ii) Late stage: Significant pathological phenotype or renal function decline occurs in adulthood (>1 month). (iii) Early and late stages: The model encompasses both cyst formation (early stage) and chronic progression (late stage).

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
