# Peer review of "Advances and Challenges in Modeling Autosomal Dominant Polycystic Kidney Disease: A Focus on Kidney Organoids"

_biomedicines, 2025, doi:10.3390/biomedicines13020523_

Round 1
Reviewer 1 Report
Comments and Suggestions for Authors
Authors described a complex field of kidney organoid research in modeling ADPKD. The review was well written and comprehensive. There were two figures in this manuscript. However, authors did not guide the readers to look at the figure in the text. For example, the figure 1 summarized three different approaches, but readers would only see the figure after reading the paragraph. Please revise your text and remind the readers to the figure in proper time.
Author Response
Comments 1:Authors described a complex field of kidney organoid research in modeling ADPKD. The review was well written and comprehensive. There were two figures in this manuscript. However, authors did not guide the readers to look at the figure in the text. For example, the figure 1 summarized three different approaches, but readers would only see the figure after reading the paragraph. Please revise your text and remind the readers to the figure in proper time.
Response1:We sincerely appreciate your valuable feedback and recognition of our work. At the same time, we apologize for the oversight of not providing explicit figure references to guide readers at appropriate points in the text. To address this issue, we have carefully revised the manuscript and incorporated direct figure citations in relevant sections to ensure a more seamless reading experience (Lines 195, 210, 257, 570,620,664,716,786,879). We greatly appreciate this insightful suggestion, as it has helped improve the overall readability and accessibility of our review.

Reviewer 2 Report
Comments and Suggestions for Authors
The manuscript titled "Advances and Challenges in Modeling Autosomal Dominant Polycystic Kidney Disease: A Focus on Kidney Organoids" reviewed the current state of organoid technology in the context of Autosomal Dominant Polycystic Kidney Disease (ADPKD). It highlights the potential of kidney organoids derived from induced pluripotent stem cells (iPSCs) for disease modeling, mechanism exploration, and drug discovery. Below are some suggestions.
1. Expand the introduction to provide a more detailed background on ADPKD, including its prevalence, clinical manifestations, and current treatment challenges.
2. Include a section that more deeply explores the molecular and cellular mechanisms underlying ADPKD, particularly focusing on how these mechanisms are recapitulated in organoid models.
3. Include a comparative analysis of organoid models with other in vitro and in vivo models used in ADPKD research.
4. Discuss the ethical implications of using iPSC-derived organoids, particularly in the context of personalized medicine and gene editing.
5. Expand the section on future directions to include emerging technologies that could further enhance organoid development, such as advanced imaging techniques, CRISPR/Cas9 advancements, and the integration of artificial intelligence for data analysis.
Author Response
Comments 1:Expand the introduction to provide a more detailed background on ADPKD, including its prevalence, clinical manifestations, and current treatment challenges.
Response 1:We sincerely appreciate your valuable suggestion to provide a more detailed background on ADPKD. In the revised manuscript, we have accordingly updated the introduction (Lines 34-44, 55-75). The revised section now includes additional information on the epidemiology of ADPKD, its clinical manifestations, the molecular mechanisms underlying disease progression, and the existing therapeutic challenges. Thank you again for your insightful feedback, which has significantly contributed to improving the clarity and depth of our manuscript.
Comments 2: Include a section that more deeply explores the molecular and cellular mechanisms underlying ADPKD, particularly focusing on how these mechanisms are recapitulated in organoid models.
Response 2:We sincerely appreciate your thorough review of our work and we apologize for not delving deeper into the molecular and cellular mechanisms of ADPKD. Based on your suggestion, we have expanded the section titled "Molecular and Cellular Mechanisms of ADPKD: Recapitulation in Organoid Models" to provide a more comprehensive exploration of these mechanisms (Lines 125-179). Through a review of relevant literature, we found that current research directly simulating the ADPKD mechanisms in organoids is still limited. Most studies focus on first simulating the pathological phenotype of ADPKD, such as cyst formation, in organoids, followed by mechanistic validation to ensure that the molecular mechanisms of the organoid models align with clinical ADPKD.
In the revised manuscript, we have detailed experimental strategies for simulating secondary insults, including pharmacological induction and environmental changes, particularly exploring the interplay between PC1/PC2 signaling, cAMP metabolism, and calcium homeostasis. Additionally, we discuss how abnormal activation of the renin-angiotensin-aldosterone system (RAAS) in organoid models is replicated, providing further evidence supporting the physiological relevance of RAAS in ADPKD pathology.
Comments 3: Include a comparative analysis of organoid models with other in vitro and in vivo models used in ADPKD research.
Response 3:We truly value your insightful suggestion to include a comparative analysis of organoid models alongside other classical in vitro and in vivo models used in ADPKD research. In response to your feedback, we have revised the manuscript and incorporated this analysis into the section "ADPKD Organoid Models: A Comparison with Classical Models" (Lines 270-376). This addition provides an overview of commonly used models in ADPKD research, highlighting their respective strengths, limitations, and comparisons with organoid models.
Comments 4. Discuss the ethical implications of using iPSC-derived organoids, particularly in the context of personalized medicine and gene editing.
Response 4:We sincerely appreciate your valuable suggestion, particularly regarding the ethical implications of iPSC-derived organoids in the context of personalized medicine and gene editing. In response to your feedback, we have revised the manuscript and added a dedicated section titled "Ethical Considerations: Unignorable Issues in ADPKD Organoids" following the discussion on organoid applications (Lines 480-512). This section highlights key ethical concerns, including patient informed consent, data privacy protection, and the responsible use of gene editing technologies in organoid research. This addition not only enhances the comprehensiveness of our review but also integrates important considerations of medical ethics and social responsibility. Once again, we truly appreciate your constructive input.
Comments 5. Expand the section on future directions to include emerging technologies that could further enhance organoid development, such as advanced imaging techniques, CRISPR/Cas9 advancements, and the integration of artificial intelligence for data analysis.
Response 5:We greatly appreciate your suggestion to expand the discussion on emerging technologies that could further advance organoid development. In response, we have revised the "Challenges and Perspectives in Autosomal Dominant Polycystic Kidney Organoid Research" section to include advancements in high-resolution imaging technologies and machine learning in the field of organoids (Lines 655-660, 670-691, 700-711, 716-761, 769-780). These technologies enable more precise characterization of organoid structure, function, and disease progression, ultimately enhancing their translational potential. We believe that these additions strengthen our discussion on the future prospects of ADPKD organoid research.

Reviewer 3 Report
Comments and Suggestions for Authors
Given the current limitations in organoid models, how reliable are they for large-scale drug screening and personalized medicine applications?
How may organoids be designed to better imitate the "second hit" concept and other genetic complexity of ADPKD?
What specific recommendations may be made to close the gap between organoid research and clinical use for ADPKD treatment?
Author Response
Comments 1: Given the current limitations in organoid models, how reliable are they for large-scale drug screening and personalized medicine applications?
Response 1:Thank you for raising this insightful question regarding the reliability of organoid models in large-scale drug screening and personalized medicine applications. We believe that organoid models have already demonstrated considerable accuracy in these areas and are emerging as a promising platform for both high-throughput drug screening and precision medicine. One of the key advantages of organoid models is their ability to capture patient-specific genetic backgrounds and 3D tissue architecture, thereby closely resembling the pathological structures and transcriptomic profiles observed in ADPKD patient-derived kidney tissues. Compared to traditional 2D cell cultures, this high degree of biomimicry enhances the predictive power of organoids in drug screening and disease modeling. As illustrated in our manuscript, several cyst inhibitors have been identified through organoid-based drug screening, which serves as strong evidence of their reliability. Furthermore, in the context of personalized medicine, patient-derived hiPSC kidney organoids may offer distinct advantages over gene-edited models, as they represent a pre-symptomatic state and have the potential to develop into fully formed kidney cysts, thereby providing a more physiologically relevant platform for studying disease progression and treatment response.
Additionally, organoids hold great promise for high-throughput screening, particularly in evaluating the efficacy and toxicity of compound libraries at a scale that traditional models cannot achieve. Compared to conventional animal models, organoids significantly reduce biological variability and batch effects, ensuring more consistent and reproducible experimental outcomes. Moreover, their use aligns with the 3R principles of animal research ethics (Replacement, Reduction, and Refinement), minimizing the reliance on animal models while maintaining experimental rigor. We appreciate your thought-provoking question, which has allowed us to further emphasize the strengths and translational potential of organoid models in ADPKD research and drug discovery.
Comments 2: How may organoids be designed to better imitate the "second hit" concept and other genetic complexity of ADPKD?
Response 2: First, we sincerely thank you for your careful review of our work and for raising this valuable question. We highly appreciate your inquiry regarding how to better simulate the “second-hit” concept and other genetic complexities in ADPKD. To improve the simulation of this phenomenon, we believe that the design of organoid models must consider genetic background, environmental factors, and intercellular interactions.
In the revised manuscript, we have added a section titled "Molecular and Cellular Mechanisms of ADPKD: Recapitulation in Organoid Models", where we expand on the methods currently used to simulate the “second-hit” phenomenon. In this section, we specifically discuss commonly used methods such as pharmacological induction (e.g., Forskolin) and changes to the organoid culture's adhesion conditions. Both of these approaches are capable of inducing cyst formation in organoids based on primary gene mutations (e.g., PKD1 or PKD2 mutations).
Regarding how to better simulate the "second-hit" phenomenon in ADPKD, we believe the following strategies could be considered:
- Gene Editing and “Second-Hit” Simulation:First, organoid models should utilize gene editing technologies (e.g., CRISPR/Cas9) to introduce primary mutations (e.g., PKD1 or PKD2 mutations) into the organoids, simulating the genetic basis of the disease. Subsequently, additional mutations could be introduced into the differentiated organoid models. Currently, technologies that can correct pathogenic PKD1 mutations in ADPKD models are available, and the use of these technologies to induce secondary mutations is theoretically feasible. However, further research is needed to determine which specific genes should be targeted for the "second-hit" mutations.
- Incorporation of Environmental Factors: Studies have shown that environmental factors such as renal ischemia-reperfusion can accelerate cyst expansion. Therefore, simulating the environmental pressures that ADPKD patients' kidneys experience (e.g., hypertension, kidney injury) could enhance the disease characteristics of the organoids. Achieving this will require more complex kidney models and a more precise simulation of the renal microenvironment. By utilizing ultra-high resolution imaging and 3D bioprinting technologies, more accurate kidney organoid models could be developed. Additionally, co-culturing organoids with endothelial cells and immune cells would facilitate vascularization and the simulation of intercellular interactions and inflammatory responses, which will help better replicate the complexity of the disease.
- High-Throughput Screening and Precise Validation: With the advancement of imaging technologies, researchers can observe the dynamic changes in organoids under various time points and stimuli. By applying machine learning and other technologies in high-throughput screening, we can identify which genetic variations or environmental factors are most likely to drive the progression of ADPKD. This would not only assist in validating the "second-hit" hypothesis but also provide new insights into understanding the molecular mechanisms of the disease.
In summary, we believe that combining these design strategies will enable future organoid models to better simulate the “second-hit” concept and genetic complexity of ADPKD, providing a more translational research platform for studying and treating the disease.
Comments 3: What specific recommendations may be made to close the gap between organoid research and clinical use for ADPKD treatment?
Response 3:We sincerely thank you for your profound questions about closing the gap between organoid research and the clinical use of ADPKD therapy. iPSC-derived kidney organoids provide unprecedented insights into the pathophysiology of ADPKD, but several challenges must be addressed in order to enhance its clinical relevance and translational potential, and in the latest manuscript we address this gap in greater detail. Here are a few key recommendations briefly summarized:
- Improve maturity and functional fidelity
Current ADPKD organoid models lack complex structure, complete nephrine maturation, and vascularization capabilities, limiting their ability to fully simulate disease. 3D bioprinting technology was used to construct organoid structures closer to pathological conditions. Promote the differentiation scheme and simulate the organoids corresponding to different disease stages; Co-culture with endothelial cells to promote vascularization, combined with perfusable microfluidic systems, may enhance its physiological relevance. In addition, optimizing the composition of the extracellular matrix (ECM) can further improve the structure and function of organoids.
- Improve the accuracy of drug detection and prediction
High-resolution imaging captures the true response of organoids to drugs. Based on machine learning, validating organoid-derived drug sensitivity against large-scale clinical trial data will be a key step. Large-scale studies that correlate in vitro organoid drug screening with patient-sourced data can improve their predictive power for personalized medicine applications.
- Standardized repetitive culture conditions
Establishing Good Manufacturing Practice (GMP) compliant protocols and automated organoid culture technologies are critical to generating reproducible, clinically relevant models at scale. The use of synthetic hydrogels with well-defined biochemical properties can further reduce batch to batch variability.

Reviewer 4 Report
Comments and Suggestions for Authors
Dear authors,
You have presented an excellent review, it is difficult to make any comments. And yet 2 questions.
1. You mention gene therapy, can you clarify which vectors can be used in gene therapy to transfer healthy donor organoids.
2. I would provide several more modern sources of literature to make the review more complete. I cannot advise you, but here are some more modern sources, perhaps they will be useful to you.
-
Advances and Challenges Toward Developing Kidney Organoids for Disease Modeling and Regenerative Medicine
Author: Melissa H. Little
Journal: Nature Reviews Nephrology
Publication Date: August 2023
Summary: This review discusses the progress in generating kidney organoids and highlights significant challenges in creating sophisticated disease models, including those for ADPKD. It also explores the potential of organoids in regenerative medicine.
Link: https://pubmed.ncbi.nlm.nih.gov/37541208/
-
Advancements in Research on Genetic Kidney Diseases Using Human Kidney Organoids
Authors: Yuki Morizane and Joseph V. Bonventre
Journal: Frontiers in Cell and Developmental Biology
Publication Date: July 2023
Summary: This article reviews the use of human kidney organoids to model genetic kidney diseases, including ADPKD. It emphasizes the utility of organoids in understanding disease mechanisms and testing potential therapies.
Link: https://www.frontiersin.org/articles/10.3389/fcell.2023.845401/full
-
Experimental Models of Polycystic Kidney Disease
Authors: John P. Calvet and Peter C. Harris
Journal: Kidney360
Publication Date: August 2023
Summary: This review examines various experimental models used in ADPKD research, including kidney organoids. It discusses their applications, advantages, and limitations in therapeutic testing.
Link: https://journals.lww.com/kidney360/fulltext/2023/08000/experimental_models_of_polycystic_kidney_disease_.23.aspx
-
Studying Kidney Diseases Using Organoid Models
Authors: Minoru Takasato and Melissa H. Little
Journal: Frontiers in Cell and Developmental Biology
Publication Date: March 2022
Summary: This paper summarizes the current kidney organoid models and discusses recent advances in modeling kidney diseases, including ADPKD. It also addresses the challenges hindering the application of organoids in disease modeling and drug evaluation.
Link: https://www.frontiersin.org/articles/10.3389/fcell.2022.845401/full
-
Disease Modeling with Kidney Organoids
Authors: Jennifer A. Lewis and Ryuji Morizane
Journal: Micromachines
Publication Date: September 2023
Summary: This article explores the use of kidney organoids as models for various diseases, including ADPKD. It highlights the strengths and weaknesses of organoid models and discusses prospective applications in disease modeling.
Link: https://www.mdpi.com/2072-666X/13/9/1384
These articles provide comprehensive insights into the advancements and ongoing challenges in modeling ADPKD using kidney organoids.
Author Response
Reviewer #4:
Comments 1: You mention gene therapy, can you clarify which vectors can be used in gene therapy to transfer healthy donor organoids.
Response 1:First, we sincerely apologize for not clearly elaborating on the selection of gene therapy vectors in our manuscript, and we truly appreciate your insightful question. In the latest version of our manuscript, we have provided a more detailed discussion on the delivery vectors used for gene editing in organoids. (Lines 450-452, 459-476)
Furthermore, we would like to emphasize that in gene modification or functional correction of iPSC-derived organoids, the selection of delivery vectors must take into account multiple factors, including gene size, delivery efficiency, tissue specificity, long-term expression capability, and safety. Different gene types may require specific vectors, and in some cases, existing vectors may need to be optimized to enhance delivery efficiency and targeting precision. However, as of now, there is no universally established standard or recommended vector specifically for organoid-based gene therapy, and research in this area remains in its exploratory phase. We look forward to future advancements in gene editing and delivery systems, which will lead to the development of more efficient and safer gene delivery strategies, ultimately facilitating the clinical translation of organoid-based gene therapies.
Comments 2: I would provide several more modern sources of literature to make the review more complete. I cannot advise you, but here are some more modern sources, perhaps they will be useful to you.
Response 2: We sincerely appreciate the five important references you provided on ADPKD and kidney organoid construction during the review process. The recommended literature not only broadened the theoretical scope of our study but also provided crucial support for the argument structure of our manuscript. Through careful review, we found that these papers offered valuable theoretical foundations and practical guidance on key issues, such as molecular regulatory mechanisms in ADPKD organoids and strategies for constructing disease animal models.
During the revision process, we carefully selected the following references for incorporation into our manuscript: Nishinakamura (2023), "Advances and Challenges Toward Developing Kidney Organoids for Clinical Applications" (Line 552); Harris, P.C. team (2023), "Experimental Models of Polycystic Kidney Disease"(Line 330); Liu, M et al. (2022), "Studying Kidney Diseases Using Organoid Models"(Line 336). We have seamlessly integrated these references into our revised manuscript, which has enhanced the precision and depth of our argumentation.
